# Challenging thermodynamics: combining immiscible elements in a single-phase nano-ceramic

Shuo Liu[1], Chaochao Dun [2,6] ✉, Qike Jiang[3], Zhengxi Xuan[1,4], Feipeng Yang [5], Jinghua Guo [5], Jeffrey J. Urban [2,6] ✉ & Mark T. Swihart [1,4,6] ✉

The Hume-Rothery rules governing solid-state miscibility limit the compositional space for new inorganic material discovery. Here, we report a non-equilibrium, one-step, and scalable flame synthesis method to overcome thermodynamic limits and incorporate immiscible elements into single phase ceramic nanoshells. Starting from prototype examples including $(NiMg)O$, $(NiAl)O_x$, and $(NiZr)O_x$, we then extend this method to a broad range of Ni-containing ceramic solid solutions, and finally to general binary combinations of elements. Furthermore, we report an "encapsulated exsolution" phenomenon observed upon reducing the metastable porous $(Ni_{0.07}Al_{0.93})O_x$ to create ultra-stable Ni nanoparticles embedded within the walls of porous $Al_2O_3$ nanoshells. This nanoconfined structure demonstrated high sintering resistance during 640 h of catalysis of $CO_2$ reforming of methane, maintaining constant 96% $CH_4$ and $CO_2$ conversion at 800 °C and dramatically outperforming conventional catalysts. Our findings could greatly expand opportunities to develop novel inorganic energy, structural, and functional materials.

Heteroatom doping of one element into another crystal lattice can introduce active sites, change electronic structure, modify coordination environment, and alter defect density. Thus, it is a general and widely used approach to optimizing material function for broad application areas such as catalysis[1], photovoltaics[2], and electrochemistry[3]. However, incompatibility of elements limits the dissolution of foreign atoms, leaving a narrow range of thermodynamically-stable solid solutions. The thermodynamic constraints are described by the Hume-Rothery rules, which predict the miscibility of solute-solvent solids based upon similarity in crystal structure, atomic radii, valency, and electronegativity. These factors vary widely for different elements (Supplementary Fig. 1), which greatly limits the compositional space for material discovery. For

example, $(NiMg)O$ solid solution-derived catalysts show excellent performance for $CO_2$ reforming of methane[4]. Unfortunately, immiscibility between NiO and other metal oxides greatly limits the development of other Ni-based catalysts by a similar route.

Conventional wet-chemistry or chemical vapor deposition methods may not provide access to certain phases of materials, making synthesis of kinetically-stable but non-equilibrium forms of matter inaccessible. For example, the miscibility gap often leads to undesirable phase segregation in the commonly adopted co-precipitation method, as solute atoms aggregate during slow nucleation and growth processes. Furthermore, fabrication of nano-ceramic solid solutions is more challenging than metal alloys due to the broader array of crystal lattices adopted by ceramics relative to simple metals. Recently,

[1]Department of Chemical and Biological Engineering, University at Buffalo, The State University of New York, Buffalo, NY 14260, USA. [2]The Molecular Foundry, Lawrence Berkeley National Laboratory, Berkeley, CA 94720, USA. [3]Instrumentation and Service Center for Physical Sciences, Westlake University, Hangzhou, Zhejiang 310024, China. [4]RENEW Institute, University at Buffalo, The State University of New York, Buffalo, NY 14260, USA. [5]Advanced Light Source, Lawrence Berkeley National Laboratory, Berkeley, CA 94720, USA. [6]These authors jointly supervised this work: Chaochao Dun, Jeffrey J. Urban, Mark T. Swihart. ✉e-mail: cdun@lbl.gov; jjurban@lbl.gov; swihart@buffalo.edu

researchers have developed some non-equilibrium synthesis strategies, such as high-temperature shockwave[5], laser scanning ablation[6], and electrical explosion[7], where the material forms in milliseconds or even nanoseconds. A fast material formation process can prevent atom diffusion and leave immiscible elements mixed in a metastable alloy phase that cannot be achieved using conventional equilibrium synthesis methods[8]. However, most of the non-equilibrium synthesis methods rely on extreme conditions, high energy input, or a specific substrate to load precursor salts and products, features that introduce barriers to scale-up.

Flame aerosol processing, dating back to prehistoric civilizations in which soot was used as a pigment, has been the most common technology for industrial production of nanoparticles such as carbon black, $TiO_2$, and fumed silica[9]. Recently, it has been extended to fabrication of advanced nanomaterials[10–13], mainly focused on modifying particle size, morphology, or crystallinity. The ability to overcome elemental thermodynamic immiscibility in such a scalable method is rare. In this research, we present a general flame aerosol strategy to integrate immiscible elements into a single nano-ceramic phase, based on a modified flame reactor[14]. We also template mesopores in $(NiAl)O_x$ solid solution nanoshells through an evaporation-driven micelle self-assembly process[15] and treat this material under reducing conditions to generate active Ni nanoparticles embedded in porous $Al_2O_3$. This resulting nanoconfined structure exhibits ultrahigh activity and stability as a catalyst for $CO_2$ reforming of methane.

## Results

### Non-equilibrium flame synthesis of ceramic solid solution nanoshells

Detailed flame synthesis procedures are described in the Methods Section (Supplementary Figs. 2, 3). The non-equilibrium synthesis concept presented mainly proceeds by a droplet-to-particle conversion process. As shown in Fig. 1, an aqueous precursor was shear-atomized into droplets of a few micrometers diameter[16,17], from which solvent evaporates in milliseconds[18,19]. As a result, the immiscible metal atoms are trapped in a metastable phase, forming a uniform ceramic solid solution. The rapid droplet-to-particle conversion tends to

generate a hollow nanoshell morphology[20,21]. The solid material formed at the droplet surface grows inwardly as water diffuses out, finally locking in the hollow sphere morphology. Similar to the quenching of alloys, rapid $N_2$-quenching prevents phase separation by avoiding the nose point of the solute partitioning TTT curve.

### Comparison between non-equilibrium (flame synthesis) and equilibrium (co-precipitation) synthesis methods

We first compare the flame synthesis (F-) with a conventional co-precipitation (CP-) method for three prototypical nickel-containing oxides: $NiO-MgO$, $NiO-Al_2O_3$, and $NiO-ZrO_2$, which are thermodynamically miscible, partially miscible and immiscible, respectively. As expected, for the $NiO-MgO$ system that satisfies the Hume-Rothery rules (Fig. 2a), both F- and CP- based methods produced homogeneous $(Ni_{0.2}Mg_{0.8})O$ solid solutions, as confirmed by high-angle annular dark-field scanning transmission electron microscopy (HAADF-STEM) with elemental mapping by energy-dispersive x-ray spectroscopy (EDS), which showed homogeneous elemental distributions of Ni and Mg (Fig. 2b, c). X-ray diffraction (XRD) patterns showed a single rock-salt phase (Supplementary Fig. 5a). For the $NiO-Al_2O_3$ system which can form a $NiAl_2O_4$ spinel phase, the CP-$NiO-Al_2O_3$ material exhibited obvious phase segregation (Fig. 2d, Supplementary Fig. 4), as reflected by a nonuniform elemental distribution (Fig. 2e) and separate $NiO$, and $NiAl_2O_4$ spinel phases, possibly accompanied by some $Al_2O_3$, in XRD (Supplementary Fig. 5b). In contrast, the flame synthesis method produced a homogeneous $F-(Ni_{0.2}Al_{0.8})O_x$ solid solution (Fig. 2f) without separated $NiO$ or $NiAl_2O_4$ phases (Supplementary Fig. 5b).

A similar phenomenon was observed for the thermodynamically immiscible $NiO-ZrO_2$ system, whose phase diagram has not been published. Conventional co-precipitation failed to dope Ni into the $ZrO_2$ lattice, instead yielding separate $NiO$ and $ZrO_2$ phases, as shown in the XRD pattern of CP-$NiO/ZrO_2$ in which characteristic $NiO$ peaks at 37.2° and 43.5° were detected (Fig. 2g, Supplementary Table 1). In contrast, the $F-(Ni_{0.2}Zr_{0.8})O_x$ exhibited a single phase without $NiO$ peaks (Fig. 2g, Supplementary Table 2). Compared to the $ZrO_2$ peaks in CP-$NiO/ZrO_2$, the slight peak shifts of $F-(Ni_{0.2}Zr_{0.8})O_x$ also demonstrated that Ni was incorporated in the $ZrO_2$ lattice, altering the cell parameter. Meanwhile, the $F-(Ni_{0.2}Zr_{0.8})O_x$ solid solution showed a homogeneous elemental distribution (Fig. 2h), while an inhomogeneous distribution of the elements was evident in CP-$NiO/ZrO_2$ (Supplementary Fig. 6). High-resolution transmission electron microscopy with fast-Fourier-transform (FFT) analysis revealed nanoscale phase segregation in CP-$NiO/ZrO_2$ (Fig. 2i). Consistent with XRD analysis, different $NiO$ and $ZrO_2$ grains were observed. In contrast, the (111) diffraction spots in $F-(Ni_{0.2}Zr_{0.8})O_x$ formed a circle of fixed diameter (Fig. 2j), indicating nanocrystalline grains of the same phase and lattice constant. These results clearly demonstrate the ability of the flame-based synthesis to overcome immiscibility predicted by violation of the Hume-Rothery rules. Synthesizing single-phase solid solutions in the $NiO-ZrO_2$ system poses a significant challenge due to the miscibility gap arising from their dramatic difference in atomic radius (1.62 vs 2.16 Å), preferred valence (+2 vs +4), electronegativity (1.91 vs 1.33), and crystal structure (rock-salt vs tetragonal). However, achieving this goal would powerfully demonstrate that the flame synthesis method can produce metastable materials regardless of elemental miscibility. Thus, we intentionally selected this prototype system to establish the concept before extending it to other Ni-based and general combinations.

In addition, materials prepared by co-precipitation exhibited a relatively high degree of crystallinity and large grains, while the flame synthesized solid solutions showed a hollow nanoshell morphology with a polycrystalline structure of nanoscale grains. These thin shells consisted of numerous grains and abundant grain boundaries, which can play critical roles in increasing performance in catalysis and related applications[22,23].

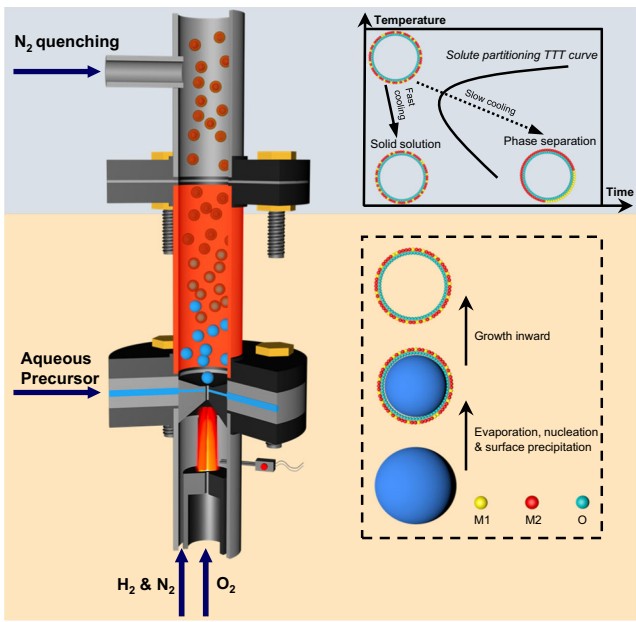

**Fig. 1 | Schematic of the non-equilibrium flame aerosol process producing ceramic solid solution nanoshells via an evaporation driven droplet-to-particle conversion followed by $N_2$ quenching.** M1 and M2 represent metal elements used in aqueous inorganic salt solutions as precursors.

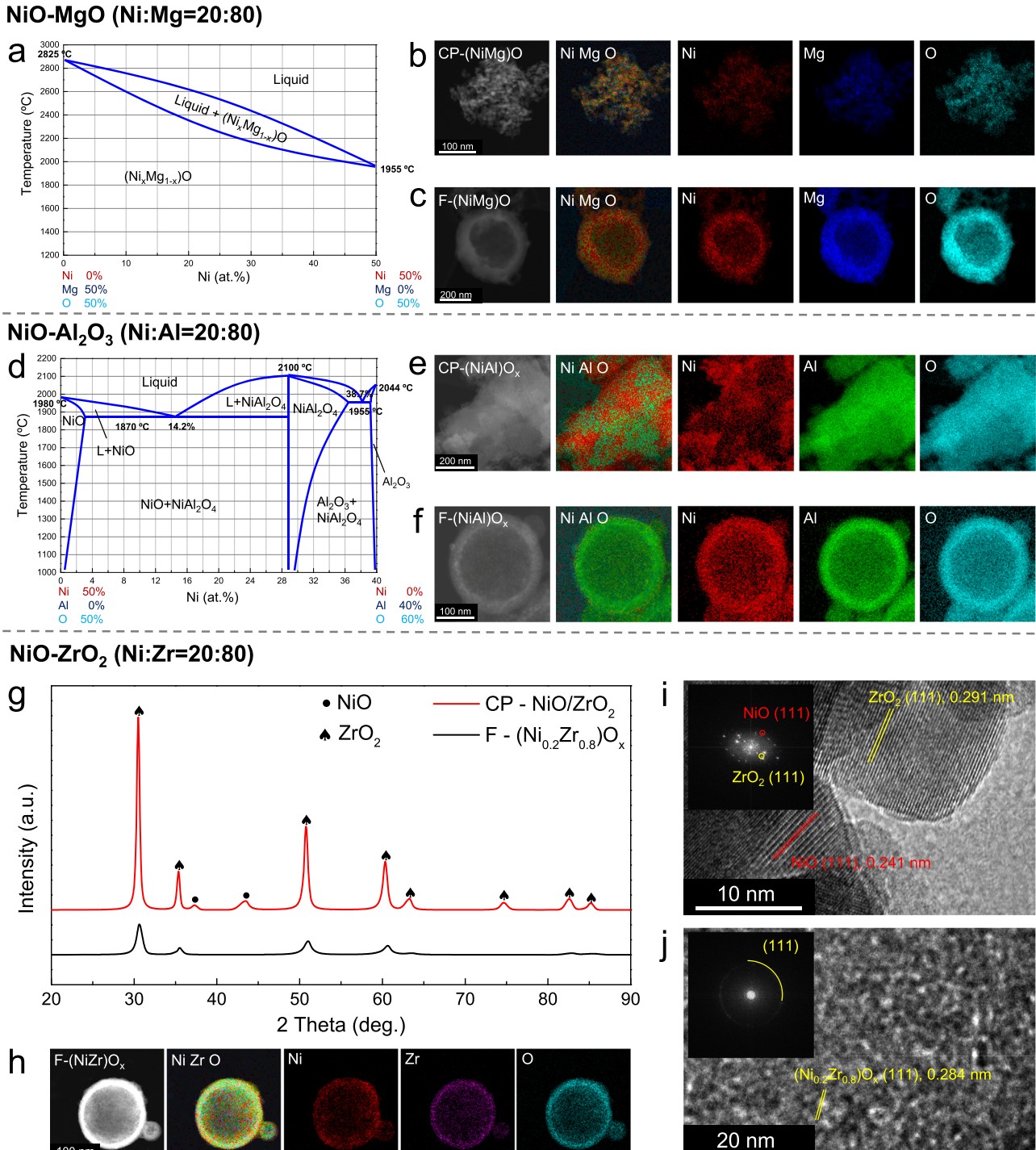

**Fig. 2 | Typical NiO-MgO, NiO-Al₂O₃, and NiO-ZrO₂ materials synthesized by co-precipitation (CP-) and flame aerosol (F-) methods, which are miscible, partly miscible, and immiscible, respectively. a** NiO-MgO phase diagram[49]; **b, c** HAADF-STEM and elemental maps of CP-(NiMg)Oₓ and F-(NiMg)Oₓ; **d** NiO-Al₂O₃ phase diagram[50]; **e, f** HAADF-STEM and elemental maps of CP-(NiAl)Oₓ and F-(NiAl)Oₓ; **g** Rietveld refined XRD patterns of CP-NiO/ZrO₂ and F-(NiZr)Oₓ; **h** HAADF-STEM and elemental maps of F-(NiZr)Oₓ; **i, j** HRTEM images and corresponding FFT patterns of CP-(NiZr)Oₓ and F-(NiZr)Oₓ.

## Extension to numerous Ni-containing ceramic solid solution nanoshells

Overcoming elemental immiscibility would greatly expand the accessible material space relative to conventional approaches (Fig. 3a). Thus, a diverse palette of Ni-containing ceramic solid solution nanoshells was created to demonstrate the generality of the non-equilibrium flame synthesis method, whose solvent metal oxides are partly miscible or immiscible with NiO (Fig. 3b). The maps exhibited homogeneous elemental distributions without any Ni aggregation

(Fig. 3c–k), and the XRD patterns of each of the Ni-containing solid solutions matched that of the solvent metal oxide, without phase separation (Supplementary Fig. 11), indicating that all materials formed homogeneous ceramic solid solution nanoshells.

For the (Ni₀.₁In₀.₉)Oₓ, smaller nanoparticles that are characteristic of formation by a gas-to-particle route were observed around the nanoshell (Supplementary Figs. 7, 9j)[11], but the Ni was still well incorporated into the In₂O₃ lattice in these nanoparticles, demonstrating that both droplet-to-particle and gas-to-particle routes in the current

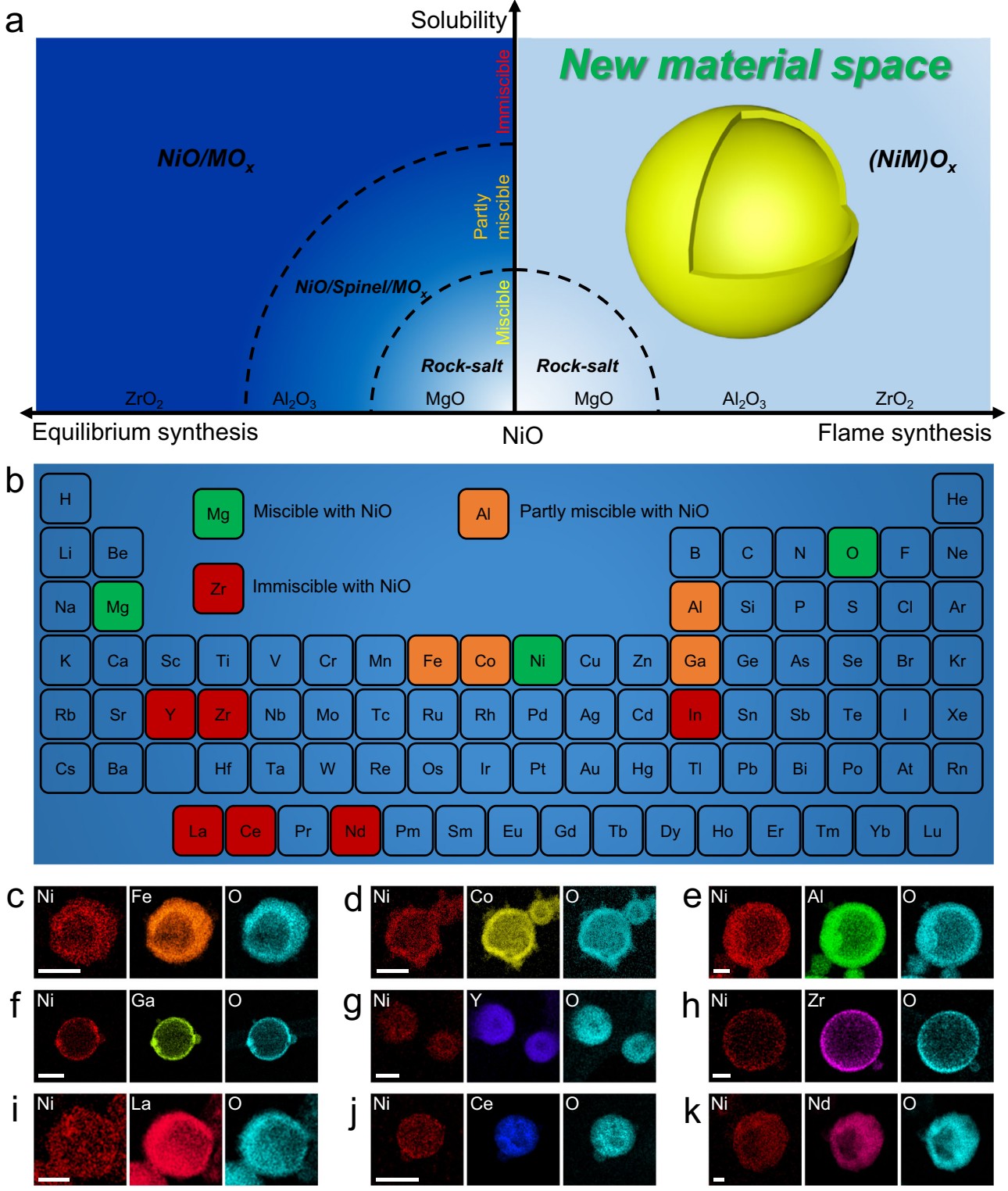

**Fig. 3 | Palette of Ni-containing ceramic solid solution nanoshells. a** Illustration of the expansion of material space, in which partly miscible and immiscible pairs exhibit phase separation in equilibrium synthesis but form a single phase using the current flame synthesis method; **b** Miscibility of NiO with other metal oxides; Elemental maps of the flame synthesized. **c** $(Ni_{0.1}Fe_{0.9})O_x$; **d** $(Ni_{0.1}Co_{0.9})O_x$; **e** $(Ni_{0.1}Al_{0.9})$ $O_x$; **f** $(Ni_{0.1}Ga_{0.9})O_x$; **g** $(Ni_{0.1}Y_{0.9})O_x$; **h** $(Ni_{0.1}Zr_{0.9})O_x$; **i** $(Ni_{0.1}La_{0.9})O_x$; **j** $(Ni_{0.1}Ce_{0.9})O_x$ and **k** $(Ni_{0.1}Nd_{0.9})O_x$, scale bars are 100 nm. Corresponding optical images, TEM images, XRD patterns, EDS spectra, and XPS profiles are shown in Supplementary Figs. 8–15.

flame aerosol process can generate homogeneous ceramic solid solutions. Detailed information related to material appearance, crystal structure, composition, and elemental state is provided in Supplementary Figs. 8–15.

**General ceramic solid solution nanoshells beyond NiO**

The successful mixing NiO into a broad range of metal oxides motivated us to further investigate the generality of this approach. As shown in Fig. 4a, to demonstrate the generality as much as

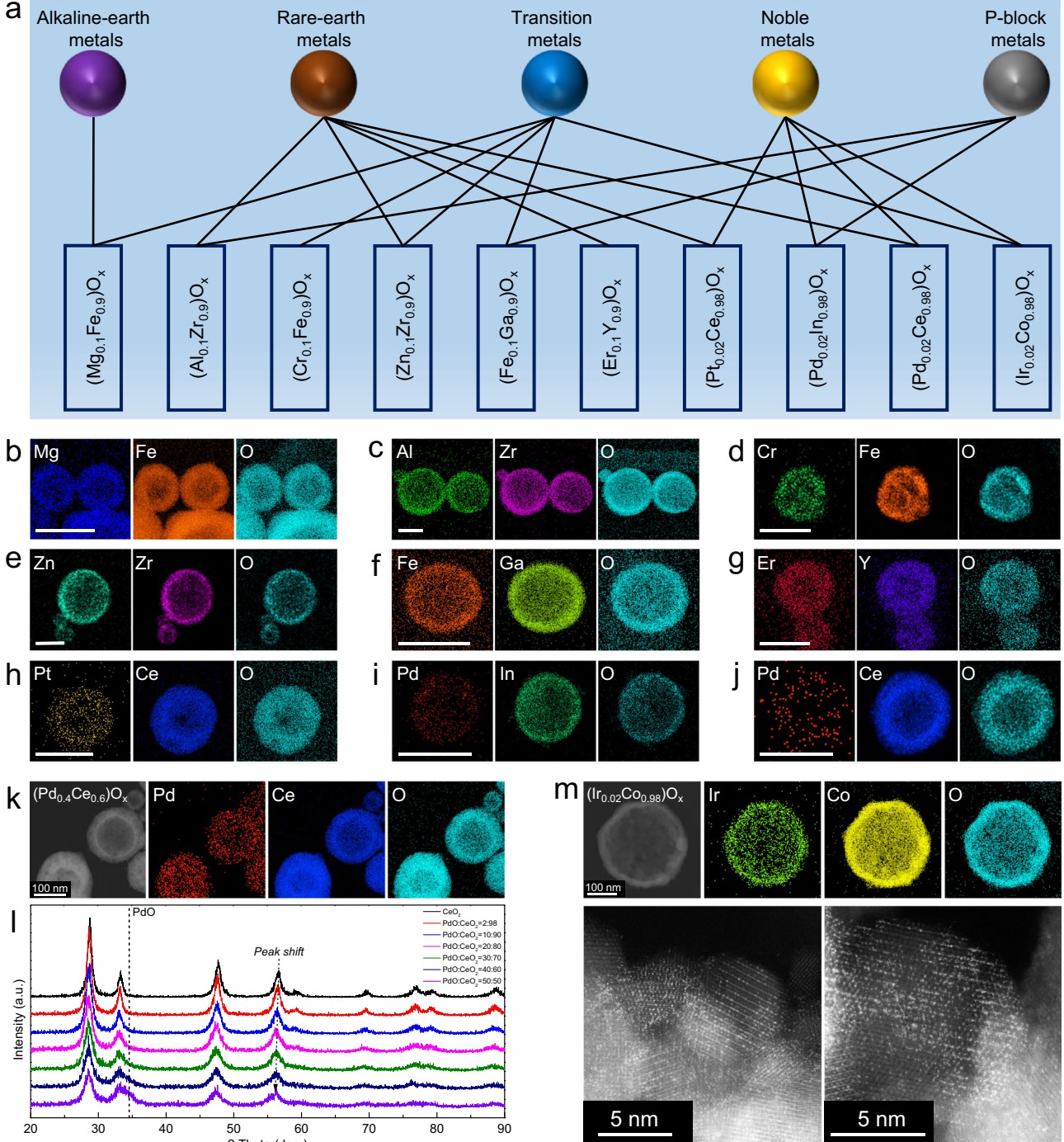

**Fig. 4 | Library of general ceramic solid solution nanoshells. a** Illustration of the generality of current flame synthesis method by combining elements from different regions of the periodic table. Elemental maps of flame synthesized (**b**) $(Mg_{0.1}Fe_{0.9})O_x$; **c** $(Al_{0.1}Zr_{0.9})O_x$; **d** $(Cr_{0.1}Fe_{0.9})O_x$; **e** $(Zn_{0.1}Zr_{0.9})O_x$; **f** $(Fe_{0.1}Ga_{0.9})O_x$; **g** $(Er_{0.1}Y_{0.9})O_x$; **h** $(Pt_{0.02}Ce_{0.98})O_x$; **i** $(Pd_{0.02}In_{0.98})O_x$ and **j** $(Pd_{0.02}Ce_{0.98})O_x$, scale bar 100 nm; **k** HAAFD-elemental maps of the $(Pd_{0.4}Ce_{0.6})O_x$; **l** XRD diffraction patterns of $(PdCe)O_x$ solid solutions with increasing Pd content from 0 to 50 mol.%; **m** HAADF-STEM elemental maps and AC-STEM of $(Ir_{0.02}Co_{0.98})O_x$. Corresponding optical images, TEM images, XRD diffraction patterns, EDS spectra, and XPS profiles are shown in Supplementary Figs. 16–21.

possible, we selected pairs of elements from different regions of the periodic table, excluding radioactive elements. The representative library of ceramic solid solution materials comprised $(MgFe)O_x$, $(AlZr)O_x$, $(CrFe)O_x$, $(ZnZr)O_x$, $(FeGa)O_x$, $(ErY)O_x$, $(PtCe)O_x$, $(PdIn)O_x$, $(PdCe)O_x$, and $(IrCo)O_x$, which are of interest for applications including heavy metal ion adsorption[24], dental filling materials[25], photoconductors[26], catalysis of $CO_2$ hydrogenation to methanol[27], dielectric material[28], up-conversion emission[29], catalysis of CO oxidation[30], $H_2$ sensors[31], catalytic methane combustion[32], and

oxygen evolution reaction under acidic conditions[33], respectively. All of them formed homogeneous ceramic solid solution nanoshell structures. STEM elemental mapping confirmed the homogeneity of all the solid solutions (Fig. 4b–j, m). XRD patterns further confirmed that the solutes were incorporated into the solvents' crystal lattices, with most particles adopting a hollow nanoshell morphology in these systems (Supplementary Figs. 17, 18). Detailed information related to material appearance, composition, and elemental state is provided in Supplementary Figs. 16 and 19–21.

We further investigated the extent to which the ratio between solute and solvent elements could be tuned. Similar to the $(NiCe)O_x$ solid solution, $PdO\text{-}CeO_2$ was chosen as a model system due to its immiscibility and broad applications in catalysis[34]. $(PdCe)O_x$ solid solution materials were synthesized with increasing Pd content from 0 to 50 mol.% in the precursor. The XRD results show only a single fluorite phase (Fig. 4l) with minimal evidence of a PdO peak, consistent with the uniform distribution of Pd, Ce, O in the $(Pd_{0.4}Ce_{0.6})O_x$ solid solution nanoshell (Fig. 4k). The slight peak shift in XRD peak position reflected incorporation of more Pd upon increasing Pd content in the precursor, accompanied by a decrease in the crystallinity after doping (Fig. 4l, Supplementary Fig. 11, Supplementary Fig. 15 and Supplementary Fig. 18). These results suggested that the atomic ratio in these systems can be readily tuned, and high solubility is likely to be achieved in ceramic solid solutions by our non-equilibrium flame synthesis. Moreover, although it is not the main focus of this work, we note that doping highly dispersed noble metal cation active sites (like Pd, Pt, Ir) into the support lattice is also feasible. To confirm it, aberration corrected scanning transmission electron microscopy (AC-STEM) revealed the crystal lattice and elemental distribution of $(Ir_{0.02}Co_{0.98})O_x$, demonstrating a high concentration of Ir active sites at the lattice positions of Co and atomically dispersed in the $Co_3O_4$ support (Fig. 4m).

## "Encapsulated exsolution" phenomenon

The $(Ni_{0.07}Al_{0.93})O_x$ was selected as a model material for further exploration. We found that the oxygen vacancy concentration of F-$(Ni_{0.07}Al_{0.93})O_x$ was twice that of CP-$(Ni_{0.07}Al_{0.93})O_x$ (Supplementary Fig. 22)[35]. For catalysis applications, we adopted an evaporation-driven micelle self-assembly method to template mesopores in the $(Ni_{0.07}Al_{0.93})O_x$ in-situ during synthesis, creating mesoporous $(Ni_{0.07}Al_{0.93})O_x$ nanoshells (Supplementary Figs. 23, 24)[15]. Using this mesoporous metastable binary oxide, we discovered an "encapsulated exsolution" phenomenon, as shown in Fig. 5. Typical exsolution behavior involves physical processes of diffusion, reduction, nucleation and growth, and generates active particles on a metal oxide surface as a reducible metal is exsolved from the oxide[36]. However, upon heating the porous $(Ni_{0.07}Al_{0.93})O_x$ solid solution in $H_2$ to reduce Ni, the pores provided internal nucleation sites within the alumina shell. Thus, the Ni diffused towards the pores in the shell rather than exclusively toward the outer surface, finally forming a structure of Ni nanoparticles encapsulated in porous and hollow $Al_2O_3$ (Fig. 5a). Transmission electron microscopy (TEM) images clearly showed that the Ni nanoparticles formed in the hollow $Al_2O_3$ after exsolution, and not primarily on the external surface (Fig. 5b). HAADF-STEM elemental mapping of the exsolved $Ni/Al_2O_3$ further demonstrated the encapsulated structure with a high concentration of Ni nanoparticles well dispersed in the hollow $Al_2O_3$ shell (Fig. 5c). We also measured soft X-ray absorption spectra (XAS) at the Ni $L_{2,3}$-edge in $(Ni_{0.07}Al_{0.93})O_x$ solid solution and reduced F-$Ni/Al_2O_3$ (Fig. 5d). The significant peak shift of $(Ni_{0.07}Al_{0.93})O_x$ compared to NiO reference indicates a substantial quantity of $Ni^{3+}$ in $(Ni_{0.07}Al_{0.93})O_x$, which reflected the high defect density and was consistent with the oxygen vacancy analysis from XPS (Supplementary Fig. 22)[37]. After reduction, the XAS profile of reduced $Ni/Al_2O_3$ overlapped with the Ni foil reference, indicating all the Ni cations were reduced to a metallic state, in agreement with Ni $2p$ XPS analysis (Supplementary Fig. 25). XRD patterns also showed the separate Ni metal phase formed after reduction, with weak peaks of $\gamma$-$Al_2O_3$ phase also appearing (Supplementary Fig. 26) while the as synthesized $(Ni_{0.07}Al_{0.93})O_x$ solid solution was amorphous.

This newly demonstrated "encapsulated exsolution" has several obvious advantages over the conventional exsolution process[36]: I. Fast exsolution speed. Exsolution of Ni nanoparticles from the current $(NiAl)O_x$ solid solution took less than 4 h at 800 °C in $H_2$ reducing atmosphere, while for exsolution of Ni from spinel, in a prior report,

most $Ni^{2+}$ remained in the spinel lattice after 7 h at 800 °C[38]. The intrinsic metastability, lattice mismatch between dopant and parent ions, and abundant oxygen vacancies lower the energy barriers to cation diffusion and phase segregation, accelerating the exsolution process[39]. II. Ultrasmall nanoparticle size. The "encapsulated exsolution" yielded small Ni nanoparticles of 8.7 nm (Fig. 5e), dramatically smaller than the Ni nanoparticles generated by conventional exsolution from various spinel and perovskite materials (>30 nm)[40–43]. III. Ultrahigh thermal stability. The nanoconfined structure after exsolution provided powerful sintering resistance. The current exsolved Ni/$Al_2O_3$ was aged at 800 °C for 30 h with only a slight increase in Ni particle size, from 8.7 to 9.7 nm based on TEM analysis (from 9.7 to 13.7 nm by XRD analysis) (Fig. 5e, Supplementary Fig. 26). This is comparable to a recent report of Pt nanoparticles encapsulated in porous $Al_2O_3$[44]. TEM images and elemental mapping demonstrated that the morphology, Ni nanoparticle dispersion, and encapsulated structure of F-$Ni/Al_2O_3$ remained the same after aging (Supplementary Fig. 27). When we increased the calcination temperature to 1000 °C, the Ni average nanoparticle size grew to 22 nm (Supplementary Fig. 28), with the majority of Ni remaining as un-sintered small Ni nanoparticles. At 1200 °C, the hollow structure was still maintained even after the Ni nanoparticles dissolved in $\alpha$-$Al_2O_3$. This demonstrated that the current "encapsulated exsolution" process can be reversed, and the material could be self-regenerative through a "exsolution $\leftrightarrows$ dissolution" cycle[45].

Current exsolution research relies on existing materials. The non-equilibrium flame aerosol process can combine nearly any pair of elements. Thus, any reducible elements (e.g., Co, Ni, Cu, Pt, Pd, Ir, Au, …) can be doped into an oxide host lattice of an irreducible element (e.g., Al, Ce, La, Mg, Zr, Y, In, …), to serve as the exsolution precursors, which dramatically expands the variety of exsolution parent materials beyond perovskites and spinels.

## A prototypical catalysis application in $CO_2$ reforming of methane

The flame synthesized and exsolved F-$Ni/Al_2O_3$ showed features that are desired in catalysis applications, including their hollow and porous structure that facilitates rapid mass transfer of reactant gases; highly dispersed active sites and abundant oxygen vacancies that increase reactant adsorption and conversion; and the nano-confined structure that impart high sintering resistance to reduce catalyst deactivation. Here, we selected $CO_2$ (dry) reforming of methane ($CH_4 + CO_2 \leftrightarrows 2H_2 + 2CO$), which converts the two most prominent greenhouse gases to a valuable chemical feedstock[46], as a representative application to examine catalytic performance (Fig. 6a).

The influence of Ni content (3, 5, 7, 10, and 15 mol.%) on catalyst performance was investigated first under continuous time-on-stream tests at 800 °C for 30 h, in which the material with 7 mol.% Ni showed the best activity and stability. Further increasing the Ni content did not improve performance (Fig. 6b–d, Supplementary Figs. 29–31). Then, the activity of the optimized F-$Ni/Al_2O_3$ catalyst was investigated at different reaction temperatures, compared to a conventional CP-$Ni/Al_2O_3$ catalyst prepared by a co-precipitation method with the same 7 mol.% Ni content (Fig. 6e–g). As expected for this endothermic reaction, $CH_4$ and $CO_2$ conversions increased with increasing reaction temperature. At low temperatures, $CH_4$ conversion was lower than $CO_2$ conversion and the $H_2/CO$ ratio was lower than 1 due to occurrence of the reverse water gas shift reaction ($CO_2 + H_2 \leftrightarrows CO + H_2O$). At 800 °C, F-$Ni/Al_2O_3$ exhibited ~96% $CH_4$ and $CO_2$ conversions with a $H_2/CO$ ratio close to 1. In general, F-$Ni/Al_2O_3$ demonstrated greater catalytic activity than CP-$Ni/Al_2O_3$.

To date, $CO_2$ reforming of methane has not reached large-scale commercial application, and the major barrier has been catalyst deactivation at high reaction temperature[47]. Thus, we investigated the stability of F-$Ni/Al_2O_3$ at 800 °C with gas hourly space velocity

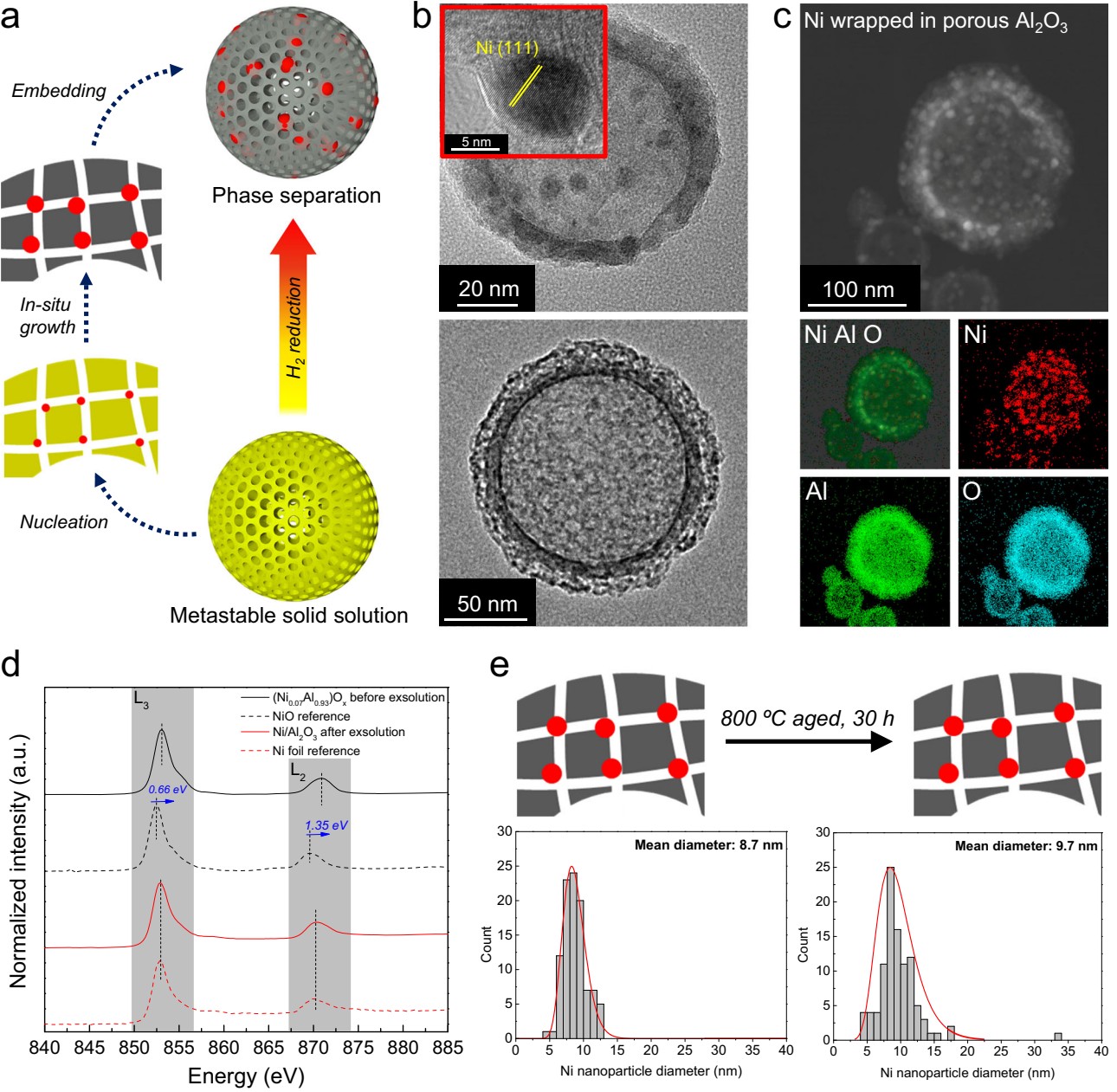

**Fig. 5 | Encapsulated exsolution behavior. a** Schematic of Ni nanoparticles exsolved from hollow and porous $(Ni_{0.07}Al_{0.93})O_x$ solid solution in $H_2$ (red-Ni, gray-$Al_2O_3$); **b** TEM images of as-synthesized $(Ni_{0.07}Al_{0.93})O_x$ and exsolved F-Ni/$Al_2O_3$; **c** HAADF-STEM and elemental maps of the exsolved Ni/$Al_2O_3$; **d** Ni L-edge XANES spectra; **e** Ni size distribution curves of Ni/$Al_2O_3$ before and after aging at 800 °C for 30 h, obtained from statistics of 100 Ni nanoparticles in TEM images (red-Ni, gray-$Al_2O_3$).

(GHSV) of 60,000 mL $g_{cat}^{-1}$ $h^{-1}$. The F-Ni/$Al_2O_3$ catalyst maintained constant ~96% $CH_4$ and $CO_2$ conversions for 640 h (27 days), with a $H_2$/CO ratio close to 1 (Fig. 6h, i). Furthermore, we increased the GHSV by a factor of 3 to 180,000 mL $g_{cat}^{-1}$ $h^{-1}$, and the F-Ni/$Al_2O_3$ catalyst still maintained the same activity and stability (Fig. 6j–l). In contrast, under the same reaction conditions, the conventional CP-Ni/$Al_2O_3$ catalyst exhibited decreasing $CH_4$ and $CO_2$ conversions, suggesting catalyst deactivation. Typically, most approaches for designing stable catalysts lead to activity-stability trade-offs, in which high stability is achieved at the expense of activity[44]. Encouragingly, the F-Ni/$Al_2O_3$ catalyst exhibited both long-term stability and ultra-high activity, demonstrating advantages of the non-equilibrium flame synthesis method compared with previously reported methods of producing Ni/$Al_2O_3$ catalysts[48].

## Discussion

This research establishes a general methodology to circumvent thermodynamic immiscibility to create a broad array of homogeneous metastable solid solution nano-ceramics. These can enable new properties and applications either in their as-synthesized metastable state, or by serving as precursors for nano-phase separated systems. This class of materials exhibits exploitable features including highly dispersed solute atoms, hollow nanoshell morphology, and high defect density, with controllable characteristics like flexible composition and solute/solvent ratio, templated pore structure, and tailored phase-separation during post-processing, enabling rational design of material properties for targeted applications, such as $CO_2$ reforming of methane. The intrinsic scalability, single-step and continuous operation, and relatively low cost of the aerosol processing approach

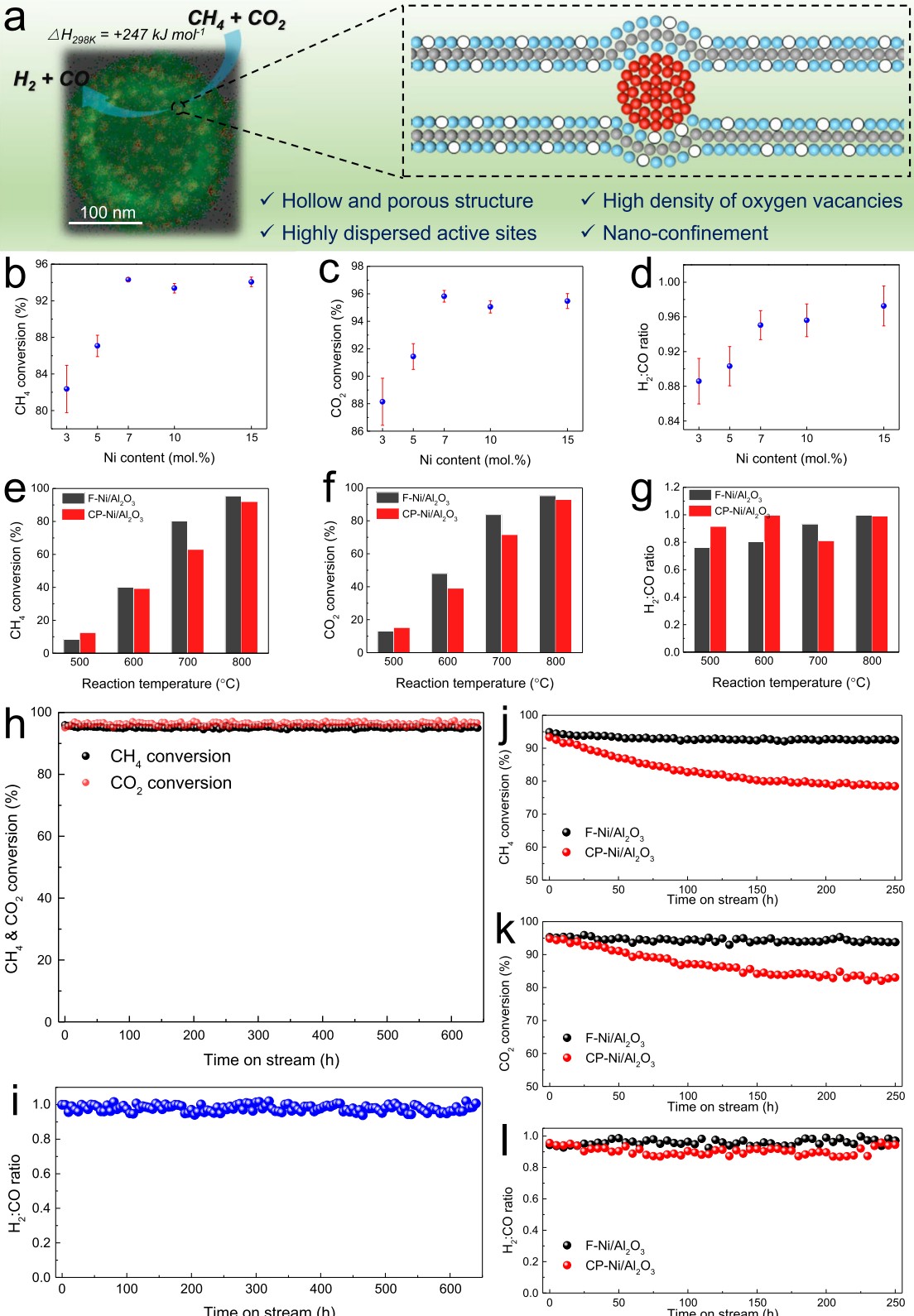

**Fig. 6 | Application of Ni/Al₂O₃ in catalysis. a** Catalyst properties and application in CO₂ reforming of methane; Activity analysis of the F-Ni/Al₂O₃ catalysts with varied Ni content. **b** CH₄ conversion. **c** CO₂ conversion, and (**d**) H₂:CO ratio at 800 °C over 30 h (Corresponding time on stream curves are shown in Supplementary Figs. 29–31; error bars represent one standard deviation of the distribution of 180 values measured over 30 h.); Activity analysis of F-Ni/Al₂O₃ and CP-Ni/Al₂O₃ catalysts at varied reaction temperatures. **e** CH₄ conversion. **f** CO₂ conversion. **g** H₂:CO ratio; Stability analysis of F-Ni/Al₂O₃ catalyst at 800 °C for 640 h. **h** CH₄ conversion and CO₂ conversion. **i** H₂:CO ratio; Stability analysis of F-Ni/Al₂O₃ and CP-Ni/Al₂O₃ catalysts at 800 °C for 250 h. **j** CH₄ conversion. **k** CO₂ conversion. **l** H₂:CO ratio. Reaction conditions in (**b**–**g** and **j**–**l**): 60 mL min⁻¹ total feed gas flow rate, CH₄:CO₂:Ar = 1:1:1, 20 mg catalyst loading, 180,000 mL g$_{cat}^{-1}$ h⁻¹ GHSV, atmospheric pressure. Reaction conditions in (**h**, **l**): 30 mL min⁻¹ total feed gas flow rate, CH₄:CO₂:Ar = 1:1:1, 30 mg catalyst loading, 60,000 mL g$_{cat}^{-1}$ h⁻¹ GHSV, atmospheric pressure.

employed here strongly suggest that it can be economically applied for large-scale production in industry, paving the way for widespread development and application of inorganic nanomaterials in fields including sensing, energy storage, and catalysis.

## Methods

### Chemicals

Precursor salts of $Ni(NO_3)_2 \cdot 6H_2O$ (99%), $Mg(NO_3)_2 \cdot 6H_2O$ (99 + %), $Al(NO_3)_3 \cdot 9H_2O$ (99 + %), $ZrO(NO_3)_2 \cdot H_2O$ (99.5%), $Co(NO_3)_2 \cdot 6H_2O$ (99 + %), $Ga(NO_3)_3 \cdot H_2O$ (99.99%), $Ce(NO_3)_3 \cdot 6H_2O$ (99.5%), and $Cr(NO_3)_3 \cdot 9H_2O$ (99%) were purchased from Acros Organics; $In(NO_3)_3 \cdot H_2O$ (99.99%), $Nd(NO_3)_3 \cdot 6H_2O$ (99.9%), $Er(NO_3)_3 \cdot 5H_2O$ (99.9%), and $Pt(NO_3)_4$ solution (Pt 15 w/w) were purchased from Alfa Asear; $Fe(NO_3)_3 \cdot 9H_2O$ was purchased from Fisher Chemical; $Zn(NO_3)_2 \cdot 6H_2O$ (98%) was purchased from Sigma Aldrich; $Pd(NO_3)_2 \cdot H_2O$ (99.8%, Pd 39% min) and $IrCl_3 \cdot 3H_2O$ (Ir 53–56%) were purchased from Thermo Scientific. NaOH pellets were purchased from Sigma Aldrich. Hexadecyltrimethylammonium bromide (CTAB, 99 + %) and HCl (37%) were purchased from Acros Organics. Ethanol (200 proof) was obtained from Decon Labs, Inc.

### Flame aerosol synthesis

A flowing gas mixture of 8.5 L min$^{-1}$ $H_2$, 15 L min$^{-1}$ $N_2$, and 7.5 L min$^{-1}$ $O_2$ was ignited to form an inverted diffusion flame. The hot combustion products passed through a converging-diverging nozzle (drill Ø0.1111 inch, #34) to form a sonic velocity hot stream. An aqueous precursor solution of desired composition was prepared by dissolving metal salts in water at a total concentration of 20 mM and desired molar ratio. When the temperature and pressure in the reactor stabilized, the liquid precursor was injected into the throat section of the nozzle (drill Ø0.1040 inch, #37) by a peristaltic pump at 300 mL h$^{-1}$ flow rate, where it was atomized by the high-velocity gas stream. These operating conditions produce a typical temperature of ~800 °C in the reactor, downstream of the nozzle. In this high temperature reaction zone, each droplet evaporated and a ceramic solid solution nanoshell formed from each droplet. Downstream of the reaction zone, the product was immediately cooled by a 140 L min$^{-1}$ $N_2$ flow to prevent phase separation and particle sintering. The product was collected on a filter membrane (Millipore Durapore PVDF, 0.22 μm nominal pore size, 29.3 cm diameter). A downstream vacuum pump provided control of pressure in the reactor system. The current laboratory-scale reactor yields roughly 1 g per hour for the materials reported here.

### Co-precipitation synthesis

The precursor solution was prepared by dissolving two inorganic salts in water with total concentration of 100 mM and desired molar ratio. Then, 1 M NaOH solution was added dropwise into the precursor solution under vigorous stirring until the pH value reached 11. The solution was stirred at 80 °C for 2 h. After that, the intermediate product was collected by vacuum filtration, washing, and drying at 100 °C overnight. Finally, the sample was calcined at 550 °C in air for 4 h. The CP-Ni/Al$_2$O$_3$ catalyst was synthesized by the same procedures with a Ni to Al molar ratio of 0.07:0.93 in the precursor. The catalyst was reduced by 20 mL min$^{-1}$ $H_2$ at 800 °C for 4 h before use as a catalyst.

### Synthesis of porous $(Ni_{0.07}Al_{0.93})O_x$ solid solution

The precursor solution was prepared by dissolving $Ni(NO_3)_2$, $Al(NO_3)_3$, and CTAB into a mixed solvent of water and ethanol. The $Al(NO_3)_3$ concentration was 30 mM with Ni to Al molar ratio of 0.07:0.93 and CTAB to Al molar ratio of 0.25:1. HCl was added to the precursor solution to reach a pH value of 2. The $H_2$ flow rate was changed to 6.5 L min$^{-1}$, which generated a temperature of ~700 °C in the reactor chamber. Other procedures were as described above for the general

flame synthesis. The intermediate product collected after flame synthesis was calcined at 550 °C in air for 4 h to remove the micelle template.

### Ni encapsulated exsolution

The porous $(Ni_{0.07}Al_{0.93})O_x$ solid solution was heated in 20 mL min$^{-1}$ $H_2$ reducing atmosphere at 800 °C for 4 h. The material after exsolution was also the F-Ni/Al$_2$O$_3$ catalyst.

### Material characterizations

High-angle annular dark-field scanning transmission electron microscopy (HAADF-STEM) imaging with elemental mapping by EDS, and live FFT patterns were obtained using a JEOL 2100-F 200 kV field-emission analytical TEM; Aberration corrected scanning transmission electron microscopy (AC-STEM) analysis was performed on a JEOL JEM ARM200F thermal-field emission microscope with a probe spherical aberration (Cs) corrector working at 200 kV; The material morphology was also characterized using a JEOL JEM 2010 TEM; XRD patterns were measured using an X-ray diffractometer (Rigaku Ultima IV) with Cu Kα source ($\lambda$ = 0.15418 nm). The diffractometer was operated at 40 mA and 40 KV in the 2θ range of 5–90° at a scan speed of 2°/min; XAS were measured at Beamline 7.3.1 at the Advanced Light Source (ALS), Lawrence Berkeley National Laboratory. This is a bending-magnet beamline with a photon energy range from 250 to 1650 eV. The base pressure of the main chamber is below $1 \times 10^{-9}$ Torr. The TEY signal was obtained by monitoring the sample drain current. All the powder samples were mounted on the sample holder using a carbon tape. To further increase the TEY intensity, silver paste was used to connect the surface of these samples to the metallic sample holder to reduce charging; X-ray photoelectron spectroscopy (XPS, Thermo Fisher, USA) measurements were conducted to analyze the surface metal state and oxygen vacancies. The photoelectron spectrometer system was configured with an Al Kα excitation source with spot size of 400 μm. Before collecting an XPS spectrum, an ion flood source was applied for charge neutralization; $N_2$ physisorption measurements (Micromeritics Tri-Star II) were used to characterize pore structure at 77 K. The samples were degassed at 250 °C for 4 h prior to analysis to remove moisture.

### Catalyst tests

Catalyst performance was measured using a continuous fixed-bed flow reactor with an internal diameter of 4 mm at atmospheric pressure. Each catalyst was loaded between quartz wool plugs in the reactor. The catalyst was heated to 800 °C under 20 mL min$^{-1}$ $H_2$ flow for 1 h and was kept at 800 °C for 4 h. Then, the reactor was switched to the targeted reaction temperature. The reactant gases ($CH_4:CO_2:Ar$ = 1:1:1) at the desired total flow rate were introduced into the reactor to conduct the dry reforming of methane reaction. The effluent gases were analyzed online by a gas chromatograph (SRI 8610) fitted with a packed column (Restek Natural Gas ShinCarbon ST) and a thermal conductivity detector (TCD). The $CH_4$ conversion, $CO_2$ conversion, and $H_2$/CO ratio were calculated based on the following Eqs. (1)–(3):

$$X_{CH_4} = \frac{F_{in,CH_4} - F_{out,CH_4}}{F_{in,CH_4}} \times 100\% \tag{1}$$

$$X_{CO_2} = \frac{F_{in,CO_2} - F_{out,CO_2}}{F_{in,CO_2}} \times 100\% \tag{2}$$

$$H_2/CO = \frac{F_{out,H_2}}{F_{out,CO}} \tag{3}$$

## Data availability

The data that supports the findings of the study are included in the main text and supplementary information files. Raw data can be obtained from the corresponding author upon request.

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

## Acknowledgements

This work was supported by the DOE Building Technology Office under Contract number DEEE-0008675 (M.T.S.); by the DOE National Energy Technology Laboratory under Grant number DE-FE0032209 (M.T.S.); and by the U.S. National Science Foundation, grant number CBET-1804996 (M.T.S.). This research used resources of the Molecular Foundry and Advanced Light Source, which is a DOE office of Science User Facility under contract no. DE-AC02-05CH11231 (J.J.U. and J.H.G.).

## Author contributions

S.L., C.C.D. and M.T.S. conceived and designed the experiments. S.L. performed the materials synthesis and catalysts measurement. S.L., C.C.D., Q.K.J., Z.X.X., F.P.Y. and J.H.G. performed the materials characterizations. S.L. wrote the first manuscript. C.C.D., J.J.U. and M.T.S. revised the manuscript. C.C.D., J.J.U. and M.T.S. jointly supervised this work.

## Competing interests

The authors declare no competing interests.
