## [Peer Review File · Nature Communications]

Challenging Thermodynamics: Combining Immiscible Elements in a Single-phase Nano-ceramicREVIEWER COMMENTS

Reviewer #1 (Remarks to the Author):

Re the Nature manuscript entitled:

“Challenging Thermodynamics: Combining Immiscible Elements in a Single-phase Nano-ceramic”

Authored by Liu et al.

The main objective of the reported research is to use flame spray pyrolysis of aqueous solutions of metal salts as a means to make materials that are not thermodynamically stable but according to the results reported here are likely kinetic products of the pyrolysis method. Fast heating and cooling are often a means to access kinetic products, so this approach is known.

In addition, by using EISA (see Brinker et al) the authors are able to produce porous shell particles which is often seen in spray pyrolysis as supersaturation occurs at surfaces driven by surface evaporation. The science is well-done and the quality of the characterization is very good. However, there are a number of issues that need attention.

It is not clear what the motivation is to study some of the binary mixtures, e.g. Ni/Zr as apparently no phase diagram is published.

Thus, the publication seems to be a compendium of kinetically stable materials that can be made that might be useful in the future. This reviewer searched for an indication of quantities of nanoparticles produced per hour and was unable to. Should be in abstract. Consequently, it is assumed that the process might work at mg/h, which is of value academically but less so from a commercial standpoint. Also, a number of the segments, e.g. NiAlO_x compositions along the NiO-Al₂O₃ tie-line including phase separation to produce Ni nanoparticles including their use as catalysts has been described in several publications not referenced:

DOI: 10.1021/cm0503026, DOI: 10.1111/jace.16632, DOI: 10.1021/acsnm.8b00847

In short, while this is a good and extensive compendium of results, in this reviewer's opinion it does not meet the level of novelty and utility that would be found in a Nature paper. On revision this paper would certainly be acceptable in other journals.

Reviewer #2 (Remarks to the Author):

The manuscript describes the oxide solid-solution possibility of the non-miscible metals with Ni. Authors have generalized this concept using different metals (M = Fe, Co, Al, Ga, Y, Zr, La, Ce, Nd). The work is novel, innovative and also important for a general audience. The Ni-Al₂O₃ catalysts obtained from FSP and co-precipitation were tested for CO₂ reforming, where flame-made particle exhibited stable 95%

conversion over the period of 250 h. The manuscript is well written and clear. However, there are few critical issues that need revision. Authors are suggested to look into those issues and revise accordingly.

1. Authors claim NiO-ZrO₂ is thermodynamically immiscible. If one looks at the data base for solid-solutions utilizing these two metals, one can find handful of those materials (ICSD 97358, ICSD 97359, ICSD 97360, ICSD 97363). Same is also true for La-Ni-O solid solution (ICSD 63398, ICSD 63396, ICSD 50244 and ICSD 44252). Authors are advised to look into all the combinations of the Ni-based solution reported in this manuscript that if they are really first time known.

2. Authors claim that the lattice distances acquired from imaging for NiO-Al₂O₃ is the arithmetic mean of the lattice distances of the phase pure NiO and Al₂O₃. This is actually misleading. The incorporation of the dopant in the parent matrix triggers the change in the cell parameter resulting to slight shifting of the XRD signals. This shift is responsible for the observed lattice distance in the imaging. Moreover, the atomic radii depend on the fold-coordination (octahedral, tetrahedral) within the crystal structure. To determine this, authors are advised to Rietveld refine their XRD patterns for conclusive data.

3. Figure 2G: What is the difference between the two XRD patterns obtained from co-precipitation and flame made NiO-ZrO₂? How does this data prove that Ni is incorporated in flame-made sample and not via co-precipitation method?

4. Figure 5, Figure S9, Figure S17: Authors have used hydrated metal salts dissolved in water as a spray solution to obtain solid solution in the flame combustion. The disadvantages of using aqueous solution could be (1) insufficient heat in the combustion vicinity (2) particles aggregate together to form large inhomogeneous particles (3) mixture of ultrafine and large particles (in the range of 100-500 nm). Looking at these figures, do the authors assume complete combustion of the aqueous droplets to form a shell-like morphology where the particles are precipitated within the droplet due to heat? The particles could actually form within such shell in high flame temperature. A similar phenomenon was also observed in ACS Appl. Mater. Interfaces 2017, 9, 37760–37777 [Figure 7 (b) and (c)].

Response to reviewer #1's comments:

Reviewer #1 (Remarks to the Author):

Re the Nature manuscript entitled:

“Challenging Thermodynamics: Combining Immiscible Elements in a Single-phase Nano-ceramic”

Authored by Liu et al.

1. The main objective of the reported research is to use flame spray pyrolysis of aqueous solutions of metal salts as a means to make materials that are not thermodynamically stable but according to the results reported here are likely kinetic products of the pyrolysis method. Fast heating and cooling are often a means to access kinetic products, so this approach is known.

Our response: Thank you for your feedback. While fast quenching is a well-known technique in the production of bulk alloy materials, creating homogeneous solid solutions in bimetallic systems with positive enthalpy of mixing poses a challenge. Developing a universal method for mixing immiscible elements in a single-phase solid solution, especially for nanomaterials, remains elusive. Existing approaches for fabricating metastable alloy nanoparticles are limited, with carbothermal shock being one of the few proposed methods (*Science Advances* **6**, no. **17** (2020): eaaz6844).

To date, a general methodology for preparing metastable nano-ceramic solid solutions is lacking, highlighting the novelty of our study. Traditional synthesis methods based on thermodynamic equilibrium face inherent limitations, as slow kinetics during nucleation and growth often lead to phase separation between immiscible components. Overcoming these thermodynamic challenges is a significant hurdle in materials science.

Our work introduces a groundbreaking non-equilibrium flame synthesis strategy, rapidly transforming liquid precursors into solid solution nanomaterials in a short-residence-time continuous process. This ultrafast process prevents phase separation by kinetically locking immiscible elements into a metastable single-phase state. Subsequent fast quenching maintains this metastable state, representing a novel approach to bypassing thermodynamic limits. This paradigm shift in ceramics preparation opens avenues for creating entirely new materials.

2. In addition, by using EISA (see Brinker et al) the authors are able to produce porous shell particles which is often seen in spray pyrolysis as supersaturation occurs at surfaces driven by surface evaporation. The science is well-done and the quality of the characterization is very good. However, there are a number of issues that need attention.

Our response: We agree that the EISA method is not novel, and our previous work (*Liu S. et al., Angewandte Chemie International Edition* 61.35, e202206870, 2022) also employed EISA for preparing mesoporous silica. In our current study, the primary focus in **Fig. 5** is not on utilizing the EISA method to produce porous $(\text{Ni}_{0.07}\text{Al}_{0.93})\text{O}_x$ solid solution; instead, it highlights the observation of encapsulated exsolution behavior during the heating of the porous $(\text{Ni}_{0.07}\text{Al}_{0.93})\text{O}_x$ solid solution in H_2 and the potential value of this new encapsulated exsolution behavior. This encapsulated exsolution behavior is reported for the first time.

3. It is not clear what the motivation is to study some of the binary mixtures, e.g. Ni/Zr as apparently no phase diagram is published.

Our response: This study aims to establish a general methodology for overcoming thermodynamic immiscibility, enabling the creation of a diverse library of homogeneous metastable solid solution nano-ceramics.

On one hand, we intentionally selected the NiO-ZrO₂ combination as a model system because of the significant immiscibility between NiO and ZrO₂ arising from the major differences in their key physicochemical properties. Specifically, ZrO₂ has a much larger cation radius (0.84 Å) compared to NiO (0.69 Å), preferred tetravalent state (+4) over divalent state (+2) of NiO, and lower electronegativity (1.33 vs 1.91). These factors lead to a huge miscibility gap according to the Hume-Rothery rules. Thus, the NiO-ZrO₂ system represents an ideal model to demonstrate the capability of the flame synthesis method to overcome thermodynamic immiscibility and achieve single-phase solid solutions. Although, to the best of our knowledge, no phase diagram for this system has been published, the materials project (<https://next-gen.materialsproject.org/>) includes 5 Ni_xZr_yO crystal structures, and all are predicted to be unstable.

On the other hand, by exploring binary mixtures like Ni/Zr, we expand the compositional space for discovering novel materials with broad applications. For instance, our work demonstrates success in electrochemistry, where incorporating Ir atoms into a Co₃O₄ lattice enhances electrocatalytic performance for the acidic oxygen evolution reaction (*Journal of the American Chemical Society* 143.13 (2021): 5201-5211). Similarly, our $(\text{Ir}_{0.02}\text{Co}_{0.98})\text{O}_x$ solid solution

exhibits a comparable structure (**Fig. 4M**). In the sensor field, our ongoing research reveals that the $(\text{Pd}_{0.02}\text{In}_{0.98})\text{O}_x$ solid solution (**Fig. 4I**) functions as a highly responsive H_2 sensor material, showing up to 97% response in just 6 seconds. Additionally, in heterogeneous catalysis, our investigation indicates that the porous $(\text{Ni}_{0.07}\text{Al}_{0.93})\text{O}_x$ solid solution serves as a catalyst with ultrahigh activity and stability for the CO_2 reforming of methane reaction (**Fig. 5, Fig. 6**). Therefore, this research could attract broad interest and provide inspiration for readers working in different fields and may lead to unexpected opportunities beyond the examples given above.

To make the motivation clearer, we also add text in the updated version:

Synthesizing single-phase solid solutions in the NiO-ZrO₂ system poses a significant challenge due to the miscibility gap arising from their dramatic difference in atomic radius (1.62 vs 2.16 Å), preferred valence (+2 vs +4), electronegativity (1.91 vs 1.33), and crystal structure (rock-salt vs. tetragonal). However, achieving this goal would powerfully demonstrate that the flame synthesis method can produce metastable materials regardless of elemental miscibility. Thus, we intentionally selected this prototype system to establish the concept before extending it to other Ni-based and general combinations.

4. Thus, the publication seems to be a compendium of kinetically stable materials that can be made that might be useful in the future. This reviewer searched for an indication of quantities of nanopowders produced per hour and was unable to. Should be in abstract. Consequently, it is assumed that the process might work at mg/h, which is of value academically but less so from a commercial standpoint.

Our response: Thank you for your suggestion. We agree that the information on production scale per hour is crucial for assessing the applicability of this method, and we also believe that this research not only holds high academic value but also carries significant industrial importance. Unlike wet chemical synthesis methods that involve numerous time-consuming steps, flame aerosol processing offers a one-step and continuous synthesis approach, making it more suitable for industrial-scale production (*Chemical Society Reviews* **45.11 (2016): 3053-3068**). Furthermore, in comparison with conventional flame spray pyrolysis techniques that mostly use organic solvents, our modified flame aerosol process utilizes aqueous precursor solutions, significantly reducing costs and eliminating environmental pollution. The yield of present laboratory-scale reactor is roughly 1 g/h, but Praxair (now Linde) has scaled thermal nozzles to gas flows of at least 2,000 times that used in our laboratory-scale reactor. Therefore,

when an industrial-scale reactor is constructed, the production yield could reach several kg/h. In fact, a different but related flame aerosol process operating at pilot scale was able to achieve a ZrO₂ nanoparticle production rate equivalent to 10 t/year as early as ten years ago (*KONA Powder and Particle Journal* **29** (2011): 251-265). We note that study found that at the 10 t/year scale, solvents and precursor comprised 45% of total production costs, providing an opportunity for our aqueous system using low-cost nitrate precursors to substantially reduce costs.

We add the yield information in the material synthesis section:

The current laboratory-scale reactor yields roughly 1 g per hour for the materials reported here.

5. Also, a number of the segments, e.g. NiAlO_x compositions along the NiO-Al₂O₃ tieline including phase separation to produce Ni nanoparticles including their use as catalysts has been de-scribed in several publications not referenced:

DOI: 10.1021/cm0503026, DOI: 10.1111/jace.16632, DOI: 10.1021/acsnm.8b00847

Our response: Thank you for your suggestion. We have cited these references in the revised manuscript to fully acknowledge the contributions of previous researchers. These references support our conclusions by highlighting distinctions between our modified flame aerosol process and conventional flame spray pyrolysis, as well as the disparities between our proposed "encapsulated exsolution" and traditional exsolution behaviors.

For example, our method in flame spray pyrolysis produced amorphous hollow and porous (NiAl)O_x solid solutions, contrasting with the methods mentioned above that yielded Ni-Al spinel crystal structures. Separating Ni nanoparticles from Ni-Al spinel, as demonstrated in in the second paper (*DOI: 10.1111/jace.16632*), resulted in incomplete exsolution, leaving most Ni in the spinel lattice. The Ni²⁺ species remaining in spinel support are presumably not catalytically active. Directly using the resulting (Ni_x(NiO)_{0.5-x}(Al₂O₃)_{0.5}) as a catalyst for CO₂ reforming of methane leads to gradual emergence of larger Ni nanoparticles on Al₂O₃ surface in reducing atmospheres. The third paper (*DOI: 10.1021/acsnm.8b00847*) illustrated this, where heating (NiO)_{0.5}(Al₂O₃)_{0.5} spinel at 1100 °C in H₂ produced 100~300 nm Ni nanoparticles. Such large nanoparticles cause a significant loss of mass activity in catalysis applications.

In contrast, the "encapsulated exsolution" approach efficiently addresses the drawbacks associated with the separation of Ni nanoparticles from the metastable, porous (NiAl)O_x solid

solution. We believe this method represents a significant advance in exsolution behavior, particularly in the following aspects:

- Fast exsolution rate. Previous exsolution start from relatively stable crystal structures, with the majority being based on perovskite or spinel materials (*ACS nano* **15.1 (2020): 81-110**). Their intrinsically stable crystal structure makes Ni exsolution difficult. As a result, numerous efforts have been made to improve the exsolution rate, such as using Fe as a guest cation to replace Ni in a perovskite (*Science Advances* **6.35 (2020): eabb1573**). The second and third papers mentioned by the reviewer also indicate that the exsolution of Ni from Ni-Al spinel is not sufficient after 7 h at 800 °C, requiring a higher temperature of 1100 °C. In contrast, in our case, the metastable nature of (NiAl)O_x solid solution facilitates the release of Ni, requiring only a 4-hour process at 800 °C.
- Small nanoparticle size. In the previous exsolution process, Ni tends to concentrate at one nucleation site and readily forms a large nanoparticle. For example, the exsolution of Ni nanoparticles from perovskite material showed an average particle size of 41 nm (*Angewandte Chemie International Edition* **60.29 (2021): 15912-15919**); exsolution from Ni-Al spinel material resulted in some 100~300 nm Ni particles as shown in the third paper mentioned by the reviewer. In our case, however, from porous (NiAl)O_x solid solution, the average Ni particle size is only 8.7 nm due to the pores providing numerous nucleation sites that can uniformly disperse the Ni.
- Nano-confined structure. The resulting Ni nanoparticles were embedded inside the porous Al₂O₃ wall but not on the surface, which provided significant thermal stability (**Fig. R1**). Similar nano-confined structures and sintering resistance also can be seen in another recent publication (*Nature materials* **21.11 (2022): 1290-1297**). In that study, Pt nanoparticles were first synthesized, and then encapsulated with porous Al₂O₃. We think our method provides an easier means of producing such an encapsulated structure.

Fig. R1. Nano-confinement structure combined with the “encapsulated exsolution” behavior.

- Abundant raw materials. Our non-equilibrium flame aerosol method can combine nearly any pair of elements. Thus, we can dope any reducible elements (e.g., Co, Ni, Cu, Pt, Pd, Ir, Au, ...) into an oxide lattice of an irreducible element (e.g., Al, Ce, La, Mg, Zr, Y, In, ...) and then conduct the exsolution process, which dramatically expands the variety of exsolution parent materials beyond perovskites and spinels considered in previous studies.

To make the novelty clearer, we revised the text in the updated version:

This newly demonstrated “encapsulated exsolution” has several advantages over the conventional exsolution process³⁸: I. Fast exsolution speed. Exsolution of Ni nanoparticles from the current (NiAl)O_x solid solution took less than 4 hours at 800 °C in H₂ reducing atmosphere, while for exsolution of Ni from spinel, in a prior report, most Ni²⁺ remained in the spinel lattice after 7 hours at 800 °C.⁴⁰ The intrinsic metastability, lattice mismatch between dopant and parent ions, and abundant oxygen vacancies lower the energy barriers to cation diffusion and phase segregation, accelerating the exsolution process.⁴¹ II. Ultrasmall nanoparticle size. The “encapsulated exsolution” yielded small Ni nanoparticles of 8.7 nm (Fig. 5E), dramatically smaller than the Ni nanoparticles generated by conventional exsolution from various spinel and perovskite materials (>30 nm).⁴²⁻⁴⁵ III. Ultrahigh thermal stability. The nanoconfined structure after exsolution provided powerful sintering resistance. The current exsolved Ni/Al₂O₃ was aged at 800 °C for 30 hours with only a slight increase in Ni particle size, from 8.7 to 9.7 nm based on TEM analysis (from 9.7 to 13.7 nm by XRD analysis) (Fig. 5E, Fig. S26). This is comparable to a recent report of Pt nanoparticles encapsulated in porous Al₂O₃.⁴⁶

Current exsolution research relies on existing materials. The non-equilibrium flame aerosol process can combine nearly any pair of elements. Thus, any reducible elements (e.g., Co, Ni, Cu, Pt, Pd, Ir, Au, ...) can be doped into an oxide host lattice of an irreducible element (e.g., Al,

Ce, La, Mg, Zr, Y, In, ...), to serve as the exsolution precursors, which dramatically expands the variety of exsolution parent materials beyond perovskite and spinel.

The papers suggested by this reviewer are added in the references:

13. Azurdia, J.A. et al. Liquid-feed flame spray pyrolysis as a method of producing mixed-metal oxide nanopowders of potential interest as catalytic materials. nanopowders along the NiO–Al₂O₃ tie line including (NiO)_{0.22}(Al₂O₃)_{0.78}, a new inverse spinel composition. *Chemistry of materials* **18**, 731-739 (2006).
40. Wang, F., Sun, K., Yi, E. & Laine, R.M. Chemical modification in and on single phase [NiO]_{0.5}[Al₂O₃]_{0.5} nanopowders produces “chocolate chip-like” Ni_x@[NiO]_{0.5-x}[Al₂O₃]_{0.5} nanocomposite nanopowders. *Journal of the American Ceramic Society* **102**, 7145-7153 (2019).
45. Liang, B. et al. Resettable Heterogeneous Catalyst:(Re) Generation and (Re) Adsorption of Ni Nanoparticles for Repeated Synthesis of Carbon Nanotubes on Ni–Al–O Thin Films. *ACS Applied Nano Materials* **1**, 5483-5492 (2018).

6. In short, while this is a good and extensive compendium of results, in this reviewer's opinion it does not meet the level of novelty and utility that would be found in a Nature paper. On revision this paper would certainly be acceptable in other journals.

Our response: We agreed that novelty and application value are key criteria for *Nature Communications*. Based on the above discussion, our work demonstrates obvious breakthrough and clear novelty compared to previous publications and related studies. Key innovations include, but are not limited to:

- For the first time, pioneering a general and scalable method for creating homogeneous nano-ceramic solid solutions with a natural hollow nanoshell morphology, templated pore structure, flexible composition and solute/solvent ratio.
- Identifying the phenomenon of "encapsulated exsolution", characterized by a rapid exsolution rate, small nanoparticle size, and remarkable thermal stability.
- Leveraging these mechanisms to develop an active and ultrastable catalyst for the CO₂ reforming of methane reaction. This catalyst maintains a consistent 96% conversion of CH₄ and CO₂ at 800 °C for over 640 hours.

Therefore, considering the highly innovative approach and the fundamentally new concepts presented, as well as the great material performance demonstrated in this work, publication in a high-impact journal like ***Nature Communications*** is appropriate, ensuring that this transformative materials synthesis strategy is brought to the attention of a broad readership. This could catalyze more research interest in exploring metastable materials and non-equilibrium synthesis techniques.

Response to reviewer #2's comments:

Reviewer #2 (Remarks to the Author):

The manuscript describes the oxide solid-solution possibility of the non-miscible metals with Ni. Authors have generalized this concept using different metals (M = Fe, Co, Al, Ga, Y, Zr, La, Ce, Nd). The work is novel, innovative and also important for a general audience. The Ni-Al₂O₃ catalysts obtained from FSP and co-precipitation were tested for CO₂ reforming, where flame-made particle exhibited stable 95% conversion over the period of 250 h. The manuscript is well written and clear. However, there are few critical issues that need revision. Authors are suggested to look into those issues and revise accordingly.

We appreciate the positive comments!

1. Authors claim NiO-ZrO₂ is thermodynamically immiscible. If one looks at the data base for solid-solutions utilizing these two metals, one can find handful of those materials (ICSD 97358, ICSD 97359, ICSD 97360, ICSD 97363). Same is also true for La-Ni-O solid solution (ICSD 63398, ICSD 63396, ICSD 50244 and ICSD 44252). Authors are advised to look into all the combinations of the Ni-based solution reported in this manuscript that if they are really first time known.

Our response: According to the Hume-Rothery rules, the thermodynamic immiscibility between NiO and ZrO₂ indicates their dramatic difference in atomic radius (1.62 vs 2.16 Å), preferred valence (+2 vs +4), electronegativity (1.91 vs 1.33), and crystal structure (rock-salt vs. tetragonal), which greatly limits the incorporation of Ni into ZrO₂ crystal lattice. Based on the reviewer's suggestion, we searched the ICSD database and find that all the XRD patterns of Ni-Zr-O materials (**Fig. R2**) are different from our (NiZr)O_x solid solution, similar to Ni-La-O (**Fig. R3**). Although, to the best of our knowledge, no phase diagram for this system has been published, the materials project (<https://next-gen.materialsproject.org/>) includes 5 Ni_xZr_yO crystal structures, and all are predicted to be unstable. These include the crystal symmetries of the ICSD entries, which all come from just 3 reports, and for which there is no clear evidence that the observed crystal structures are in thermodynamically stable states.

In fact, the aim of this study is not to discover new crystal structures, instead, it is to combine immiscible elements to fabricate new metastable ceramic solid solutions that are not accessible in conventional synthesis methods. Therefore, in our study, the XRD patterns of all the as-

synthesized binary solid solutions always remain that of the solvent metal oxide. For example, NiO and MgO are thermodynamically miscible and (NiMg)O solid solution is easy to prepare, but NiO is immiscible with most metal oxides and other NiO-based solid solutions are difficult to prepare (despite formation of some stable or metastable spinel and perovskite phases in some cases). The non-equilibrium flame synthesis method in this study can overcome elemental immiscibility and dope a larger amount of Ni into any metal oxide crystal lattice. We further extend this concept to nearly any pair of elements in the periodic table and can achieve a very high solubility (e.g., more than 40% Pd in CeO₂ shown in **Fig. 4I**). Therefore, this fundamentally new concept of circumventing thermodynamic limitations can greatly expand the space of solid solution materials with broad applications in various fields.

Fig. R2. Calculated XRD patterns of the suggested Ni-Zr-O materials in ICSD database.

Fig. R3. Calculated XRD patterns of the suggested Ni-La-O materials in ICSD database.

To clarify this issue, we revised the text of this part as follows:

The maps exhibited homogeneous elemental distributions without any Ni aggregation observed (**Fig. 3C-K**), and the XRD patterns of each of the Ni-containing solid solutions matched that of the solvent metal oxide, without phase separation (**Fig. S11**), indicating that all materials formed homogeneous ceramic solid solution nanoshells.

2. Authors claim that the lattice distances acquired from imaging for NiO-Al₂O₃ is the arithmetic mean of the lattice distances of the phase pure NiO and Al₂O₃. This is actually misleading. The incorporation of the dopant in the parent matrix triggers the change in the cell parameter resulting to slight shifting of the XRD signals. This shift is responsible for the observed lattice distance in the imaging. Moreover, the atomic radii depend on the fold-coordination (octahedral, tetrahedral) within the crystal structure. To determine this, authors are advised to Rietveld refine their XRD patterns for conclusive data.

Our response: Thanks for pointing this out. The material we discussed here is NiO-ZrO₂. Indeed, incorporation of Ni in the ZrO₂ lattice matrix leads to slight peaks shift of the XRD signals and the change of lattice distance. Based on the reviewer's suggestion, we have removed the previous misleading statement. We also refined the XRD patterns of the CP-NiO/ZrO₂ and F-(Ni_{0.2}Zr_{0.8})O_x. We have revised the text as follows:

Conventional co-precipitation failed to dope Ni into the ZrO₂ lattice, instead yielding separate NiO and ZrO₂ phases, as shown in the XRD pattern of CP-NiO/ZrO₂ in which characteristic NiO peaks at 37.2° and 43.5° were detected (Fig. 2G, Table S1). In contrast, the F-(Ni_{0.2}Zr_{0.8})O_x exhibited a single phase without NiO peaks (Fig. 2G, Table S2). Compared to the ZrO₂ peaks in CP-NiO/ZrO₂, the slight peak shifts of F-(Ni_{0.2}Zr_{0.8})O_x also demonstrated that Ni was incorporated in the ZrO₂ matrix lattice, altering the cell parameter. Meanwhile, the F-(Ni_{0.2}Zr_{0.8})O_x solid solution showed a homogeneous elemental distribution (Fig. 2H), while an inhomogeneous distribution of the elements was evident in CP-NiO/ZrO₂ (Fig. S6).

The Rietveld refined XRD patterns are shown in Fig. 2G:

Fig. 2G. Rietveld refined XRD patterns of CP-NiO/ZrO₂ and F-(NiZr)O_x;

Detailed peak information has been added in Table S1 and Table S2, also shown below. As can be seen there, the small peak shift in the flame-synthesized material corresponds to about a 0.4% decrease in d -spacing. We also note that the crystallite size is smaller for the flame-synthesized material.

Table S1. Rietveld refined XRD peak information of CP-NiO/ZrO₂ shown in Figure 2G.

No.	2-theta (deg)	d (Å)	Height (cps)	FWHM (deg)	Size (Å)
1	30.47	2.931	885.7	0.41	209
2	35.35	2.537	176.9	0.41	213
3	37.25	2.412	19.9	0.80	110
4	43.48	2.079	39.6	1.24	72
5	50.73	1.798	396.6	0.50	184
6	60.29	1.534	230.1	0.59	162
7	63.30	1.468	49.2	0.92	106
8	74.56	1.272	34.0	0.92	114
9	82.52	1.168	50.1	0.93	119
10	85.06	1.140	33.2	0.73	154

Table S2. Rietveld refined XRD peak information of F-(Ni_{0.2}Zr_{0.8})O_x shown in Figure 2G.

No.	2-theta (deg)	d (Å)	Height (cps)	FWHM (deg)	Size (Å)
1	30.60	2.919	136.6	0.82	105
2	35.44	2.531	31.1	0.69	127
3	50.99	1.790	61.7	1.03	90
4	60.62	1.526	40.6	1.05	91
5	63.55	1.463	6.5	1.53	64
6	82.81	1.165	6.7	1.46	76
7	85.48	1.135	4.7	2.49	45

3. Figure 2G: What is the difference between the two XRD patterns obtained from co-precipitation and flame made NiO-ZrO₂? How does this data prove that Ni is incorporated in flame-made sample and not via co-precipitation method?

Our response: As shown in **Fig 2G**, **Table S1** and **Table S2**, NiO peaks can be observed in CP-NiO/ZrO₂, but no NiO peaks appear in F-(Ni_{0.2}Zr_{0.8})O_x. The slight peak shift of F-(Ni_{0.2}Zr_{0.8})O_x also demonstrates that the Ni was incorporated in the ZrO₂ lattice. Meanwhile, by comparing the elemental mappings of CP-NiO/ZrO₂ and F-(Ni_{0.2}Zr_{0.8})O_x (**Fig. S6**, **Fig. 2H**), we can see that the distribution of elements in CP-NiO/ZrO₂ appears non-uniform, whereas F-(Ni_{0.2}Zr_{0.8})O_x exhibits homogeneous distribution. Also, the FFT pattern of CP-NiO/ZrO₂ shows separated NiO and ZrO₂ phases, while the FFT pattern of F-(Ni_{0.2}Zr_{0.8})O_x shows a single phase (**Fig 2I, J**). In our opinion, taken together, all of these characterization results convincingly show that the Ni was incorporated into the ZrO₂ lattice in the flame-made sample but not by the co-precipitation method, where it formed a separate NiO phase.

4. Figure 5, Figure S9, Figure S17: Authors have used hydrated metal salts dissolved in water as a spray solution to obtain solid solution in the flame combustion. The disadvantages of using aqueous solution could be (1) insufficient heat in the combustion vicinity (2) particles aggregate together to form large inhomogeneous particles (3) mixture of ultrafine and large particles (in the range of 100-500 nm). Looking at these figures, do the authors assume complete combustion of the aqueous droplets to form a shell-like morphology where the particles are precipitated within the droplet due to heat? The particles could actually form within such shell in high flame temperature. A similar phenomenon was also observed in ACS Appl. Mater. Interfaces 2017, 9, 37760–37777 [Figure 7 (b) and (c)].

Our response: These issues involve the intrinsic difference between our modified flame aerosol process and conventional flame spray pyrolysis (FSP). The disadvantages of using aqueous precursors can largely be overcome by our method. As shown in **Fig.1**, our flame reactor separates the flame and material formation regions by a converging–diverging nozzle. This contrasts with conventional FSP in which the precursor solution is injected directly into the flame, and in which the solvent is often the primary fuel for the flame. In our system, hydrogen is the fuel, and combustion is complete before the precursor enters the system. Therefore, in the droplet-to-particle conversion pathway in our system, the material formation mechanism is droplet evaporation but not droplet combustion. This produces a hollow nanoshell morphology because the evaporation happens at the droplet surface, producing a solute concentration that

is maximum at the surface. Thus, the solid material formed on the droplet surface and then grew toward the center, finally locking in a hollow sphere structure. The flame provides the energy for evaporation and heating of the water, resulting in a much lower synthesis temperature than in conventional FSP. This, in turn, means that less energy must be removed to achieve rapid quenching. Conventional FSP cannot operate at the particle synthesis temperatures used here (at or below $\sim 800^{\circ}\text{C}$) because the combustion will not be stable at those temperatures.

Therefore, although the particle size is larger than the conventional FSP-prepared material, the hollow nanoshell morphology can greatly increase the surface area, and we can template pores in the shell to further increase the surface area. For example, in our previous study (*Liu S. et al., Angewandte Chemie International Edition* 61.35 (2022): e202206870.), the porous silica prepared in our flame synthesis method had a BET surface area exceeding $1000\text{ m}^2/\text{g}$, which is dramatically higher than silica nanoparticle prepared by conventional FSP with BET surface area $\sim 100\text{ m}^2/\text{g}$. Our method allows for flexible design of material structures according to specific applications, by adjusting precursor composition and flow rate, reaction temperature and pressure, as well as a post-synthesis phase-separation process, etc. For example, for catalysis application, the small active nanoparticle size is key. The Ni nanoparticles exsolved from our (NiAl) O_x solid solution have an average size of 8.7 nm and show great catalytic performance for CO_2 reforming of methane, while the Ni nanoparticles exsolved from traditional FSP prepared NiAl_2O_4 spinel show 100~300 nm particle size (DOI: 10.1021/acsanm.8b00847). In addition, in our ongoing study, we have developed a method to introduce solid materials into the precursor solution as a support, which can greatly decrease the particle size below 5 nm (Fig. R4).

The advantages of using aqueous precursors are also evident, such as low cost and environmental friendliness.

Fig R4. TEM image and particle size distribution of metal nanoparticles on carbon support material prepared by the flame aerosol process, derived from our ongoing work that is not included in this manuscript.

REVIEWERS' COMMENTS

Reviewer #1 (Remarks to the Author):

The authors have successfully answered almost all this reviewer's concerns save one. The only one that we are in disagreement is the rates of production given that the authors report just 1g/h. Then they argue that commercial entities have scaled to much greater. While this may be true, TiO₂, SiO₂, ZrO₂ are all prepared by FSP of volatile metal chlorides at multi-ton quantities per year. Nanocerox produces kg/h but burns solvent as fuel. One might argue, any rate is good...but the difference between 1g/h and kg/h is really really significant. The cost of using water which must first be vaporized means very very expensive use of gaseous fuel. This reviewer finds this argument insufficient and still recommends a lesser journal but the work is of high quality.

Reviewer #2 (Remarks to the Author):

The revised manuscript reports solid-solution of non-miscible solids with Ni as CO₂ reforming catalysts. Authors have well rebutted and revised their manuscript based the comments from both the reviewers. The revised manuscript will have lots of impact both in the material designing strategy and catalytic performance. Hence I would accept this manuscript in the present form.

Final revision

Reviewer #1 (Remarks to the Author):

The authors have successfully answered almost all this reviewer's concerns save one. The only one that we are in disagreement is the rates of production given that the authors report just 1g/h. Then they argue that commercial entities have scaled to much greater. While this may be true, TiO₂, SiO₂, ZrO₂ are all prepared by FSP of volatile metal chlorides at multi-ton quantities per year. Nanocerox produces kg/h but burns solvent as fuel. One might argue, any rate is good...but the difference between 1g/h and kg/h is really really significant. The cost of using water which must first be vaporized means very very expensive use of gaseous fuel. This reviewer finds this argument insufficient and still recommends a lesser journal but the work is of high quality.

Our response: We are happy to address your concerns and thank you for your positive feedback on the quality of this work. 1 g/h is the yield of lab-scale reactor, which is not low for nanomaterials preparation in the lab, and the scalability of FSP is well-known. Meanwhile, we hold the view that using water as a solvent is cost-effective compared to organic solvents like ethanol. In industrial production, the H₂ can be replaced by natural gas, which could further decrease the cost. Nonetheless, the scale-up of our specific method for oxide nanoparticle synthesis remains to be demonstrated. Thus, we have altered the text in the main manuscript from “The intrinsic scalability, single-step and continuous operation, and relatively low cost of the aerosol processing approach employed here make it ideal for large-scale production in industry....” to “The intrinsic scalability, single-step and continuous operation, and relatively low cost of the aerosol processing approach employed strongly suggest that it can be economically applied for large-scale production in industry,”

Reviewer #2 (Remarks to the Author):

The revised manuscript reports solid-solution of non-miscible solids with Ni as CO₂ reforming catalysts. Authors have well rebutted and revised their manuscript based the comments from both the reviewers. The revised manuscript will have lots of impact both in the material designing strategy and catalytic performance. Hence I would accept this manuscript in the present form.

Our response: We sincerely thank you for your favor and support of this work!